# Rethinking Homogeneity of Vision and Text Tokens in Large Vision-and-Language Models

## Abstract

Large vision-and-language models (LVLMs) typically treat visual and textual embeddings as homogeneous inputs to a large language model (LLM). However, these inputs are inherently different: visual inputs are multi-dimensional and contextually rich, often pre-encoded by models like CLIP, while textual inputs lack this structure. In this paper, we propose Decomposed Attention (D-Attn), a novel method that processes visual and textual embeddings differently by decomposing the 1-D causal self-attention in LVLMs. After the attention decomposition, D-Attn diagonalizes visual-to-visual self-attention, reducing computation from $\mathcal{O}(|V|^2)$ to $\mathcal{O}(|V|)$ for $|V|$ visual embeddings without compromising performance. Moreover, D-Attn debiases positional encodings in textual-to-visual cross-attention, further enhancing visual understanding. Finally, we introduce an $\alpha$-weighting strategy to merge visual and textual information, maximally preserving the pre-trained LLM's capabilities with minimal modifications. Extensive experiments and rigorous analyses validate the effectiveness of D-Attn, demonstrating significant improvements on multiple image benchmarks while significantly reducing computational costs. Code, data, and models will be publicly available.

## 1 Introduction

Large Vision-and-Language Models (LVLMs) (Liu et al., 2024b) have become pivotal in advancing artificial intelligence, enabling models to understand multimodal content by integrating visual and textual information. These models have shown significant advancements in various applications, such as image captioning, visual question answering, and multi-modal assistant, marking a substantial leap forward in cross-modal reasoning. By leveraging the strengths of pre-trained large language models (LLMs) like LLaMA (Touvron et al., 2023; Zheng et al., 2023) and Mistral (Jiang et al., 2023), and powerful visual encoders such as CLIP (Radford et al., 2021), LVLMs are pushing the boundaries of cross-modal understanding, making AI more capable of interpreting and reasoning about complex, real-world scenarios.

In most state-of-the-art LVLMs (Liu et al., 2024b; Li et al., 2024a; Tong et al., 2024a), visual inputs are processed within an LLM in the same manner as textual inputs. Specifically, visual inputs are first encoded by a pre-trained visual encoder, such as CLIP, into a sequence of visual embeddings. These embeddings are then passed through a lightweight adapter layer and concatenated with textual embeddings derived from the text prompts. The concatenated visual and textual embeddings are treated as homogeneous input embeddings and subsequently fed into a pre-trained LLM. In this approach, visual and textual embeddings are treated and processed uniformly.

In this paper, we rethink the homogeneity of visual and textual tokens in LVLMs and challenge this conventional paradigm:

*"Visual and textual inputs are **created different**, and thus we propose to **process them differently** within a large vision-and-language model."*

It is evident that visual and textual embeddings are created different. Visual embeddings are derived by passing one or more two-dimensional images through a visual encoder, while textual embeddings are generated through a lookup of learnable parameters from a one-dimensional sequence of text token IDs. These distinctions introduce significant differences in the information encoded within

each type of embedding, which necessitates different modeling and processing strategies within the LLM. Key distinctions include:

• Visual embeddings inherently encode contextual information from all other visual embeddings, whereas textual embeddings lack such intrinsic contextual awareness of other textual tokens.

• Visual inputs are intrinsically multi-dimensional (*e.g.* images are 2-D). Concatenating visual and textual embeddings into a 1-D sequence and processing them in a causal, language-centric manner can introduce undesirable modeling biases.

To address these challenges, we introduce Decomposed Attention, or D-Attn, a novel framework designed to handle visual inputs more efficiently and effectively in LVLMs. In Section 2.1, we first demonstrate that the causal self-attention mechanism (Vaswani, 2017) in an LVLM can be decomposed into three components: (1) visual-to-visual self-attention (V2V Self-Attn), (2) textual-to-visual cross-attention (T2V Cross-Attn), and (3) textual-to-textual self-attention (T2T Self-Attn), as illustrated in Figure 1. By leveraging this decomposition, we concentrate on the vision-related components, specifically the V2V Self-Attn and T2V Cross-Attn, while addressing how to effectively merge the T2V and T2T attentions.

In Section 2.2, we argue that since each visual embedding inherently encodes contextual information about other visual embeddings, it is redundant to relearn this information within the LVLM. Therefore, we propose diagonalizing the V2V Self-Attn, significantly reducing the computational complexity from $\mathcal{O}(|V|^2)$ to $\mathcal{O}(|V|)$ for $|V|$ visual embeddings without sacrificing performance. This optimization is particularly advantageous when processing high-resolution images or long video sequences.

In Section 2.3, we identify an undesirable positional bias that arises from concatenating visual and textual embeddings into a 1-D sequence. To address this issue, we propose to debias T2V Cross-Attn by removing rotary/relative positional encodings within T2V Cross-Attn. Notably, this modification, though seemingly straightforward, is difficult to implement in conventional LVLMs without our proposed attention decomposition framework.

Finally, in Section 2.4, we derive an $\alpha$-weighting strategy to merge the visual information from T2V Cross-Attn with the textual information from T2T Self-Attn. The $\alpha$-weighting approach is analytically equivalent to the inherent attention operations within LVLMs, introducing minimal architectural changes and requiring no additional learnable parameters. This ensures that the pre-trained LLM retains its full capability for competitive downstream performance.

In summary, our proposed D-Attn not only reduces computational complexity but also outperforms its self-attention counterparts by a significant margin. Under fair comparisons, D-Attn is able to process 8x more visual embeddings or train 5x faster, consistently outperforming its self-attention counterpart across a range of image benchmarks. We conduct rigorous ablation studies to validate the effectiveness of our V2V Diagonal-Attn, debiased positional encodings, and $\alpha$-weighting strategies. Furthermore, we develop D-Attn using open-source models and train it on publicly available datasets to ensure reproducibility. Code, data, and models will be made publicly available.

## 2 DECOMPOSED ATTENTION

### 2.1 BACKGROUND AND OVERVIEW

As discussed in Section 1, visual and textual inputs are created different, and therefore we propose to process them differently within an LVLM. We begin by decomposing the causal self-attention mechanism in an LVLM when both visual and textual embeddings are present. As illustrated in Figure 1, causal self-attention can be split into three distinct components: (1) visual-to-visual self-attention (V2V Self-Attn), (2) textual-to-visual cross-attention (T2V Cross-Attn), and (3) textual-to-textual self-attention (T2T Self-Attn). Together, these attention components form the foundation for processing and integrating visual information in LVLMs:

• **V2V Self-Attn** captures contextual relationships between visual embeddings by allowing each visual embedding to attend to other visual embeddings.

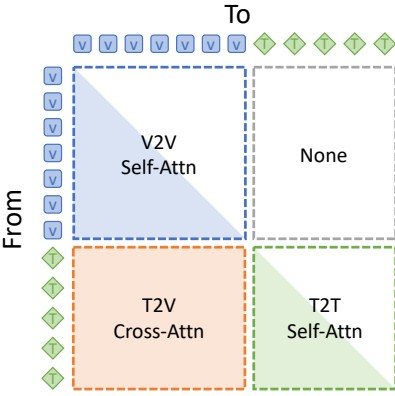

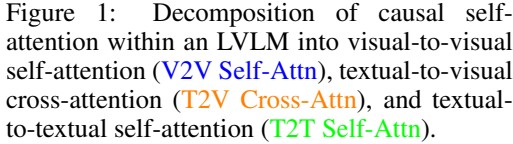

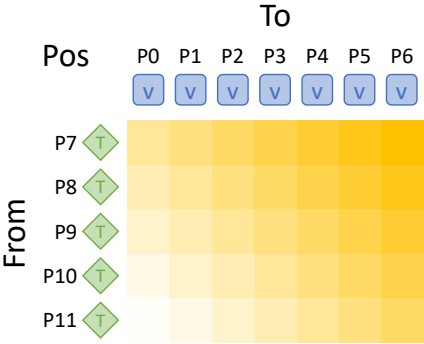

Figure 1: Decomposition of causal self-attention within an LVLM into visual-to-visual self-attention (V2V Self-Attn), textual-to-visual cross-attention (T2V Cross-Attn), and textual-to-textual self-attention (T2T Self-Attn).

Figure 2: Positional bias in T2V Cross-Attn arising from the concatenation of visual and textual embeddings into a 1-D sequence and the resulting rotary/relative positional encodings. Embeddings further away have lower values (lighter color).

• **T2V Cross-Attn** gathers visual information by allowing textual embeddings to attend to visual embeddings.

• **Weighted combination** of T2V Cross-Attn and T2T Self-Attn merges visual and textual information into the textual embeddings.

Since T2T Self-Attn operates similarly to standard attention in LLMs, we leave it unchanged and focus instead on the challenges unique to handling visual embeddings in LVLMs. With the attention decomposition, we can easily manipulate and enhance these vision-related aspects of LVLMs. In Section 2.2, we propose diagonalizing the V2V Self-Attn, significantly reducing the computational complexity from $\mathcal{O}(|V|^2)$ to $\mathcal{O}(|V|)$ for $|V|$ visual embeddings without compromising performance. In Section 2.2, we propose removing rotary/relative positional encodings within T2V Cross-Attn to mitigate undesirable positional bias between visual and textual embeddings. Lastly, in Section 2.4, we derive an $\alpha$-weighting strategy for merging T2V Cross-Attn and T2T Self-Attn, introducing minimal changes and thus preserving the pre-trained LLM's capability for competitive downstream performance.

## 2.2 V2V ATTENTION

In LVLMs, V2V Self-Attn is used to model the contextual relationships between visual embeddings. Given that visual embeddings are created by passing visual inputs through a pre-trained encoder (such as CLIP ViT (Dosovitskiy, 2020)), each visual embedding already encapsulates contextual information from other visual embeddings. This insight suggests that relearning these contextual relationships through self-attention in the LVLM may be redundant. To address this redundancy, we propose to diagonalize V2V Self-Attn, where each visual embedding attends only to itself, rather than to all other visual embeddings. Specifically, for visual embeddings $V \in |V| \times d$:

$$\bar{V} = \mathrm{SA}(V,V) = \mathtt{fc}_o \left( \underbrace{\mathrm{softmax}\left( \frac{\mathtt{fc}_q(V)\ \mathtt{fc}_k(V)^T}{\sqrt{d}}, \mathtt{dim}=1 \right)}_{\text{diagonalize}} \mathtt{fc}_v(V) \right) \tag{1}$$

$$\Rightarrow \mathtt{fc}_o\left( \mathbb{1}\ \mathtt{fc}_v(V) \right) = \mathtt{fc}_o\left( \mathtt{fc}_v(V) \right) \tag{2}$$

, where $\mathbb{1}$ is an identity matrix of size $|V| \times |V|$, and $\mathtt{fc}_q$, $\mathtt{fc}_k$, $\mathtt{fc}_v$, and $\mathtt{fc}_o$ are standard fully connected layers in an attention module for query, key, value, and output, respectively.

By turning the softmax attention matrix into an identity matrix, we essentially force each visual embedding to only attend to itself, bypassing the need for pairwise interactions between visual embeddings. As shown in Equation 2 and Figure 3b, this diagonalization simplifies the self-attention

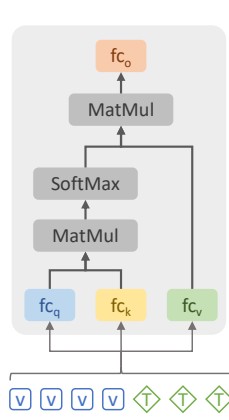
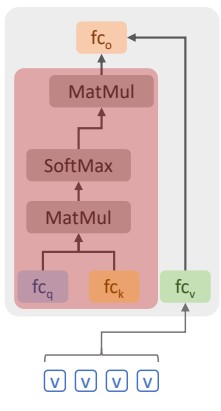
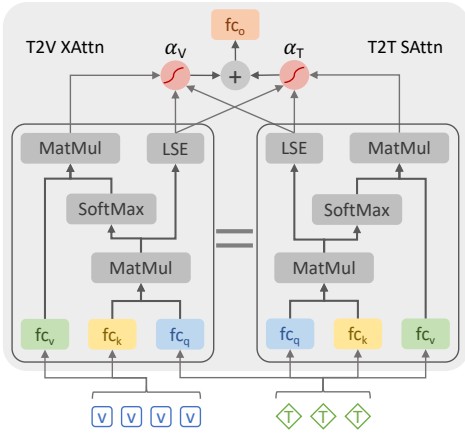

(a) Conventional attention operations.

(b) Diagonalized V2V attention.

(c) Proposed $\alpha$-weighting strategy to merge T2V Cross-Attn and T2T Self-Attn.

Figure 3: Module architecture and operations of (a) conventional attention in LVLM with visual and texual embeddings concatenated as a homogeneous input sequence, (b) V2V Diagonal-Attn, where the expensive computation of softmax attention weight is skipped, and (c) $\alpha$-weighting strategy to merge T2V Cross-Attn and T2T Self-Attn equivalent to LVLM's inherent attention operations for retaining a pre-trained LLM's full capability.

operation to only two fully connected layers, thus significantly reducing the computational complexity from $\mathcal{O}(|V|^2)$ to $\mathcal{O}(|V|)$ for $|V|$ visual embeddings. V2V Diagonal-Attn is particularly valuable when dealing with high-resolution images or long video inputs, where the number of visual embeddings $|V|$ becomes large. Notably, in our experiments, we demonstrate that this method achieves similar performance to full attention while offering significant computational savings.

## 2.3 T2V ATTENTION

In T2V Cross-Attn, textual embeddings interact with visual embeddings to incorporate visual information. To align with an LVLM's native self-attention operations and architecture, unlike previous methods that add separate cross-attention modules (Alayrac et al., 2022), we reuse and share the existing weights from the LVLM's self-attention, modifying only the query, key, and value assignments, and revising the attention mask to be non-causal. As shown in Figure 3c left, textual embeddings are used as the query, while visual embeddings serve as the key and value.

Additionally, we observe a significant issue with positional bias in T2V Cross-Attn if we follow the exact attention operation in LVLMs. When visual and textual embeddings are concatenated into a single 1-D sequence, an example of the positional IDs for textual and visual embeddings is given in Figure 2. We can see that the rotary/relative positional encodings skew attention weights based on the positional distance between visual and textual embeddings. For example, distant pairs such as the textual embedding at P11 and visual embedding at P0 receive lower attention weight than pairs closer together, like the textual embedding at P7 and visual embedding at P6. This bias can hinder effective vision-language interaction for tasks requiring a comprehensive understanding of visual context.

To address this issue, we propose to debiase T2V Cross-Attn by discarding the rotary/relative positional encodings within, effectively setting the relative positional differences to zero. Notably, this modification is challenging to implement in conventional LVLMs but becomes straightforward with our decomposed T2V Cross-Attn and T2T Self-Attn framework. To compensate for this removal, we introduce learnable positional encodings, similar to those used in CLIP, to the visual embeddings before they are passed into the LLM.

## 2.4 $\alpha$-WEIGHTING

Once visual information is gathered via T2V Cross-Attn and textual information via T2T Self-Attn, the next challenge is how to effectively merge these two streams of information. Existing methods often cascade T2V Cross-Attn and T2T Self-Attn (Alayrac et al., 2022) or introduce learnable tanh/sigmoid gates (Alayrac et al., 2022). These approaches involve significant architectural changes or introduce additional parameters, which can break the integrity and degrade the performance of pre-trained LLMs.

Instead, we propose an $\alpha$-weighting strategy for merging the T2V and T2T attentions, analytically derived from the original LVLM attention formulation. This approach introduces no additional parameters and retains equivalence with conventional LVLM attention, thereby preserving the pre-trained LLM's capabilities. For a textual query $t$, its attention to textual and visual embeddings can be expressed as:

$$\bar{t} = \texttt{Attn}(t, [V, T]) = \sum_i^L \frac{e^{\boldsymbol{q}_t \cdot \boldsymbol{k}_i}}{\sum_l^L e^{\boldsymbol{q}_t \cdot \boldsymbol{k}_l}} \boldsymbol{v}_i \tag{3}$$

, where $\boldsymbol{q}, \boldsymbol{k}, \boldsymbol{v}$ are projected query, key, value within an attention module, respectively. $\boldsymbol{k}$ and $\boldsymbol{v}$ are projected from the concatenated visual and textual embeddings $[V, T]$. For $N$ visual embeddings and $M$ textual embeddings, $\boldsymbol{k}_i \in \{\boldsymbol{k}_{v_1}, ..., \boldsymbol{k}_{v_N}, \boldsymbol{k}_{t_1}, ..., \boldsymbol{k}_{t_M}, \}$, where $\boldsymbol{k}_{v_j}$ and $\boldsymbol{k}_{t_l}$ represent the key corresponding to the $j$-th visual embedding and $l$-th textual embeddings, respectively. Similarly $\boldsymbol{v}_i \in \{\boldsymbol{v}_{v_1}, ..., \boldsymbol{v}_{v_N}, \boldsymbol{v}_{t_1}, ..., \boldsymbol{v}_{t_M}\}$. We then rewrite Equation 3 by splitting key value from $V$ and from $T$:

$$\sum_i^L \frac{e^{\boldsymbol{q}_t \cdot \boldsymbol{k}_i}}{\sum_l^L e^{\boldsymbol{q}_t \cdot \boldsymbol{k}_l}} \boldsymbol{v}_i = \sum_i^N \frac{e^{\boldsymbol{q}_t \cdot \boldsymbol{k}_{v_i}}}{\sum_l^L e^{\boldsymbol{q}_t \cdot \boldsymbol{k}_l}} \boldsymbol{v}_{v_i} + \sum_i^M \frac{e^{\boldsymbol{q}_t \cdot \boldsymbol{k}_{t_i}}}{\sum_l^L e^{\boldsymbol{q}_t \cdot \boldsymbol{k}_l}} \boldsymbol{v}_{t_i} \tag{4}$$

$$= \frac{\sum_n^N e^{\boldsymbol{q}_t \cdot \boldsymbol{k}_{v_n}}}{\sum_l^L e^{\boldsymbol{q}_t \cdot \boldsymbol{k}_l}} \sum_i^N \frac{e^{\boldsymbol{q}_t \cdot \boldsymbol{k}_{v_i}}}{\sum_n^N e^{\boldsymbol{q}_t \cdot \boldsymbol{k}_{v_n}}} \boldsymbol{v}_{v_i} + \frac{\sum_m^M e^{\boldsymbol{q}_t \cdot \boldsymbol{k}_{t_m}}}{\sum_l^L e^{\boldsymbol{q}_t \cdot \boldsymbol{k}_l}} \sum_i^M \frac{e^{\boldsymbol{q}_t \cdot \boldsymbol{k}_{t_i}}}{\sum_m^M e^{\boldsymbol{q}_t \cdot \boldsymbol{k}_{t_m}}} \boldsymbol{v}_{t_i} \tag{5}$$

$$= \frac{\sum_n^N e^{\boldsymbol{q}_t \cdot \boldsymbol{k}_{v_n}}}{\sum_l^L e^{\boldsymbol{q}_t \cdot \boldsymbol{k}_l}} \texttt{XA}(t, V) + \frac{\sum_m^M e^{\boldsymbol{q}_t \cdot \boldsymbol{k}_{t_m}}}{\sum_l^L e^{\boldsymbol{q}_t \cdot \boldsymbol{k}_l}} \texttt{SA}(t, T) \tag{6}$$

$$\equiv \alpha_V \texttt{XA}(t, I) + \alpha_T \texttt{SA}(t, T) \tag{7}$$

For numerical stability, modern deep learning packages take log of the summed exponentials:

$$\text{Let } S_V = \log \left( \sum_n^N e^{\boldsymbol{q}_t \cdot \boldsymbol{k}_{v_n}} \right), \text{ and } S_T = \log \left( \sum_m^M e^{\boldsymbol{q}_t \cdot \boldsymbol{k}_{t_m}} \right) \tag{8}$$

Then the weights $\alpha_V$ can be expressed as:

$$\alpha_V = \frac{\sum_n^N e^{\boldsymbol{q}_t \cdot \boldsymbol{k}_{v_n}}}{\sum_l^L e^{\boldsymbol{q}_t \cdot \boldsymbol{k}_l}} = \frac{e^{S_V}}{e^{S_V} + e^{S_T}} = \frac{1}{1 + e^{-(S_V - S_T)}} = \texttt{Sigmoid}(S_V - S_T) \tag{9}$$

We can similarly derive that $\alpha_T = \texttt{Sigmoid}(S_T - S_V) = 1 - \alpha_V$.

In summary, to merge visual information from T2V Cross-Attn and textual information from T2T Self-Attn while retaining equivalence to original attention in an LVLM, we propose $\alpha$ weighting, a weighted sum strategy with weights $\alpha_V$ and $\alpha_T$ analytically derived in Equation 9. As shown in Figure 3c, $\alpha$ weighting introduces no additional parameters and minimal architectural/operational changes and retains equivalence with the native LVLM attention, thereby retaining a pre-trained LLM's full capability and outperforming alternative merging strategies in our experiments.

## 3 EXPERIMENTS

### 3.1 IMPLEMENTATION DETAILS

**Model:** Our proposed D-Attn model is built based on the architecture of LLaVA (Liu et al., 2024b). It is constructed using three primary components: a pre-trained SigLip (Zhai et al., 2023) visual encoder, a randomly initialized two-layer MLP adapter with RMSNorm (Zhang & Sennrich, 2019), and a pre-trained LLM. We modify only the decoder layer and self-attention mechanisms within the LLM to implement our D-Attn. In this paper, we experiment with two different LLM families: Mistral v0.3 7B (Jiang et al., 2023), and Gemma 2 9B (Team et al., 2024).

**Training:** The training of D-Attn follows a three-stage strategy outlined in ShareGPT4V (Chen et al., 2023). In the first stage, the MLP adapter is pre-trained on LLaVA's LAION/CC/SBU(Liu et al., 2024b; Schuhmann et al., 2022; Sharma et al., 2018; Ordonez et al., 2011) 58k for modality alignment. In the second stage, the entire model is fine-tuned using 1.25M dense captions from the ShareGPT4V-PT dataset (Chen et al., 2023). In the third and final stage, we perform instruction tuning using a combined dataset of 665k examples from LLaVA-1.5 (Liu et al., 2024b) and 102k dense captions from ShareGPT4V (Chen et al., 2023). The entire training procedure completes in under 24 hours on 32 H100 GPUs. Detailed hyperparameters are provided in the Appendix.

**Evaluation:** Following LLaVA's evaluation protocol, we evaluate D-Attn on ten image benchmarks, including VQA-v2 (Goyal et al., 2017), GQA (Hudson & Manning, 2019), SQA-I (Lu et al., 2022), VQA-T (Mao et al., 2016), MME (Fu et al., 2024), MMB (Liu et al., 2023), SEED-I (Li et al., 2023), LLaVA-W (Liu et al., 2024b), MMVP (Tong et al., 2024b), and MMStar (Chen et al., 2024).

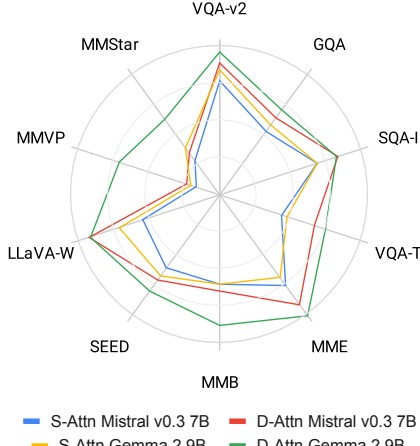

Figure 4: Performance comparison between proposed D-Attn models and their self-attention (S-Attn) counterparts on a range of popular image benchmarks. Detailed results are available in Table 1.

Our primary objective is not to achieve state-of-the-art performance but to rigorously validate the effectiveness of our proposed D-Attn framework. To ensure fair comparisons and facilitate reproducibility, we train D-Attn using only publicly available datasets through supervised fine-tuning and construct the model with open-source pre-trained LLMs and visual encoders. For stronger performance, researchers may scale up training data and models or apply more advanced training techniques such as Reinforcement Learning from Human Feedback (RLHF)(Bai et al., 2022) or Direct Preference Optimization (DPO)(Rafailov et al., 2024), which we leave as future work.

### 3.2 MAIN RESULTS

As illustrated in Figure 4, our D-Attn models consistently outperform their self-attention (S-Attn) counterparts across a range of image benchmarks. We conduct experiments using Gemma 2 9B and Mistral v0.3 7B LLMs. To ensure a fair comparison, both D-Attn and S-Attn models are trained on the same datasets using identical training strategies and are constructed with the same pre-trained visual encoders and LLMs. This experiment validates the effectiveness of the proposed D-Attn framework. In Section 3.3 and Table 2, we further demonstrate that D-Attn offers significant computational advantages over its S-Attn counterpart. Specifically, by employing the V2V Diagonal-Attention mechanism, we reduce the computational complexity from $\mathcal{O}(|V|^2)$ to $\mathcal{O}(|V|)$ for $|V|$ visual embeddings.

Table 1 presents the results of our D-Attn models and their S-Attn counterparts alongside other state-of-the-art LVLMs on ten popular image benchmarks. For reference, we include models such as Instruction BLIP (Dai et al., 2023), BLIP3 (Xue et al., 2024), VILA (Lin et al., 2024), IDEFICS (Laurençon et al., 2024a), Mini-Gemini (Li et al., 2024c), Cambrian (Tong et al., 2024a), Qwen-VL / Qwen2-VL (Wang et al., 2024), Intern-XC / Intern-XC 2.5 (Zhang et al., 2024), CuMo (Li et al., 2024b), and LLaVA-1.5 / LLaVA-1.6 (Liu et al., 2024b;a). When compared with

Table 1: Main results on a range of popular image benchmarks for our D-Attn models, their S-Attn counterparts, and other SoTA models.

| Method | LLM | Data | VQA-v2 | GQA | SQA-I | VQA-T | MME | MMB | SEED-I | LLaVA-W | MMVP | MMStar |
|--------|-----|------|--------|-----|-------|-------|-----|-----|--------|---------|------|--------|
| InstructBLIP | Vicuna 7B | 130.2M | - | 49.2 | 60.5 | 50.1 | - | 36.0 | 60.5 | 60.9 | - | - |
| BLIP 3 | Phi 3 3.8B | 3T | - | - | 88.3 | 71.0 | 1288.0 | 76.8 | 72.2 | - | - | 48.1 |
| VILA | LLaMA 2 7B | 51M | 79.9 | 62.3 | 68.2 | 64.4 | 1533.0 | 68.9 | 61.1 | 69.7 | - | - |
| IDEFICS | LLaMA 7B | 354M | 50.9 | 38.4 | - | 25.9 | - | 48.2 | - | - | - | - |
| Mini-Gemini | LLaMA 3 8B | 9.5M | - | 64.5 | 75.1 | 70.2 | 1606.0 | 65.8 | 73.7 | - | 18.7 | - |
| Cambrian | LLaMA 3 8B | 9.5M | - | 64.6 | 80.4 | 71.7 | 1547.1 | 75.9 | 74.7 | - | 51.5 | - |
| Qwen-VL | Qwen 7B | 1.4B | 78.8 | 59.3 | 67.1 | 63.8 | - | 38.2 | 56.3 | - | - | - |
| Qwen2-VL | Qwen 2 7B | UNK | - | - | - | 84.3 | - | 83.0 | - | - | - | 60.7 |
| Intern-XC | InternLM 7B | 1.1B | - | - | - | - | 1528.4 | 74.4 | 66.9 | - | - | - |
| Intern-XC 2.5 | InternLM 2 7B | UNK | - | - | - | 78.2 | - | 82.2 | 75.4 | - | - | 59.9 |
| CuMo | Mistral 0.2 7B | 2.9M | 82.2 | 64.9 | 73.9 | 67.0 | 1548.6 | 73.0 | 72.1 | 85.7 | - | - |
| LLaVA-1.5 | Vicuna 1.5 7B | 1.2M | 78.5 | 62.0 | 66.8 | 58.2 | 1510.7 | 64.3 | 66.1 | 63.4 | 20.0 | 32.8 |
| LLaVA-1.6 | Mistral 0.2 7B | 1.4M | 82.2 | 64.8 | 72.8 | 65.7 | 1498.0 | 68.7 | 72.2 | 83.2 | 32.0 | 36.1 |
| S-Attn | Mistral 0.3 7B | 2.5M | 80.3 | 61.8 | 72.7 | 62.2 | 1533.1 | 70.3 | 70.5 | 70.7 | 28.0 | 36.8 |
| D-Attn | Mistral 0.3 7B | 2.5M | 82.9 | 64.4 | 75.7 | 68.3 | 1598.6 | 71.3 | 72.6 | 79.8 | 30.0 | 38.3 |
| S-Attn | Gemma 2 9B | 2.5M | 81.8 | 63.0 | 72.8 | 63.2 | 1506.6 | 70.3 | 71.9 | 74.6 | 29.3 | 39.2 |
| D-Attn | Gemma 2 9B | 2.5M | 84.3 | 65.9 | 75.5 | 70.7 | 1636.7 | 76.5 | 74.6 | 79.7 | 45.3 | 45.0 |

Table 2: Ablations on V2V Diagonal-Attn and debiased positional encodings.

| Diag. Attn | Debiased Pos. | max $|V| \uparrow$ | sec / it $\downarrow$ | GQA | VQA-T | MME | MMB | SEED-I | LLaVA-W | MMStar |
|------------|---------------|-------------------|----------------------|-----|-------|-----|-----|--------|---------|--------|
| N | N | 9k | 11.25 | 61.8 | 62.2 | 1533.1 | 70.3 | 70.5 | 70.7 | 36.8 |
| Y | N | **74k** | **2.24** | 63.4 | 63.4 | 1507.6 | 68.8 | 70.7 | 71.2 | 32.6 |
| Y | Y | **74k** | **2.24** | **64.4** | **68.3** | **1598.6** | **71.3** | **72.6** | **79.8** | **38.3** |

other SoTA models, our D-Attn models achieve competitive performance, despite being trained on much fewer and publicly available data only, and using a simple supervised fine-tuning training strategy.

Lastly, we present qualitative comparisons between our D-Attn model and its S-Attn counterpart in Figure 5. We observe that the D-Attn model provides answers that are more faithful to the input image and offers more visual details compared to the S-Attn model. For example, in the first figure illustrating snowboarding and skiing, D-Attn effectively distinguishes between the two activities, accurately identifying one person as skiing and the other as snowboarding. While in the fourth Diamond Head figure, D-Attn provides more details about the scene such as "encircled by a road that winds its way around the base", and "Beyond the crater, the city of Honolulu sprawls out".

## 3.3 ABLATIONS AND ANALYSES

We first conduct ablation studies on the V2V Diagonal-Attn, as detailed in Table 2. To demonstrate the computational advantages, we measure the maximum number of visual embeddings ($|V|$) that an LVLM can process during training before encountering a GPU out-of-memory error. We also record the training speed in seconds per iteration (sec/it) with the same $|V|$. As shown in Table 2, by diagonalizing the V2V Self-Attn, our model can process up to 8 times more visual embeddings or train up to 5 times faster. While additional optimization techniques such as FlashAttention (Dao et al., 2022), DeepSpeed (Rasley et al., 2020), or Megatron (Shoeybi et al., 2019) can further improve memory and speed, they are orthogonal to our V2V Diagonal-Attn and still fundamentally have a computational complexity of $\mathcal{O}(|V|^2)$ for the V2V attention. In terms of performance, V2V Diagonal-Attn performs comparably to conventional LVLMs across various benchmarks, supporting our hypothesis that visual embeddings have already encoded contextual information, obviating the need for re-learning via the LLM's Self-Attn.

Next, we perform an ablation study on debiased positional encodings, also reported in Table 2. By debiasing the T2V Cross-Attn, our D-Attn model achieves consistent performance improvements over models with biased positional encodings across multiple image benchmarks. This modification cannot be easily implemented in conventional LVLMs but is rather straightforward with our proposed attention decomposition, and it brings no additional computational costs.

Table 3: Ablations on various strategies for merging visual and textual tokens.

| Merging Strategy | #Params | GQA | SQA-I | VQA-T | MME | MMB | SEED-I | LLaVA-W | MMStar |
|---|---|---|---|---|---|---|---|---|---|
| Cascade | 9.0 B | 64.1 | 72.9 | 67.0 | 1586.1 | 71.1 | 71.8 | 76.0 | 36.4 |
| Tanh | **7.6 B** | 56.6 | 73.4 | 50.0 | 1337.6 | 62.4 | 59.4 | 59.3 | 33.3 |
| Sigmoid | **7.6 B** | **64.7** | 71.9 | 66.8 | 1548.1 | 69.2 | 72.4 | 73.2 | 35.8 |
| $\alpha$-weighting (ours) | **7.6 B** | 64.4 | **75.7** | **68.3** | **1598.6** | **71.3** | **72.6** | **79.8** | **38.3** |

Furthermore, we experiment with different merging strategies in Table 3, including (1) **Cascade**, where the T2V Cross-Attn module is decoupled and cascaded with T2T Self-Attn; (2) **Tanh**, where T2V Cross-Attn is weighted by a learnable tanh gate and then summed with T2T Self-Attn; (3) **Sigmoid**, where T2V Cross-Attn and T2T Self-Attn are weighted summed with learnable gates $\sigma$ and $1 - \sigma$, respectively; and (4) $\alpha$-**weighting** strategy proposed in this paper. As shown in Table 3, our $\alpha$-weighting strategy achieves superior performance compared to other strategies without introducing additional parameters like the cascade strategy. Since $\alpha$-weighting introduces minimal architectural and operational changes to an LLM's self-attention module, it maximally retains the LLM's pre-trained capabilities, likely leading to superior fine-tuning performance on downstream tasks.

Table 4: Detailed scores for MME (Fu et al., 2024).

| Model | existence | count | position | color | posters | celebrity | scene | landmark | artwork | OCR |
|---|---|---|---|---|---|---|---|---|---|---|
| S-Attn | 190.0 | 165.0 | 121.7 | 180.0 | 134.4 | 161.2 | 166.3 | 157.8 | 128.0 | 102.5 |
| D-Attn | 195.0 | 170.0 | 143.3 | 195.0 | 161.6 | 172.6 | 163.0 | 166.8 | 137.0 | 132.5 |

Table 5: Detailed scores for SEED (Li et al., 2023).

| Model | Scene Understanding | Instance Identity | Instance Location | Instance Attributes | Instances Counting | Spatial Relation | Instance Interaction | Visual Reasoning | Text Understanding |
|---|---|---|---|---|---|---|---|---|---|
| S-Attn | 76.9 | 74.5 | 74.7 | 67.3 | 64.2 | 57.8 | 73.2 | 76.1 | 44.7 |
| D-Attn | 78.1 | 78.2 | 77.5 | 68.6 | 67.5 | 61.0 | 73.2 | 80.9 | 65.8 |

Table 6: Detailed scores for MMB (Liu et al., 2023).

| Model | action recognition | attribute recognition | celebrity recognition | function reasoning | nature relation | object localization | ocr | social relation | spatial relationship | struct. img-txt understanding |
|---|---|---|---|---|---|---|---|---|---|---|
| S-Attn | 88.8 | 83.7 | 78.7 | 74.6 | 70.8 | 50.6 | 66.6 | 83.7 | 28.8 | 33.3 |
| D-Attn | 90.7 | 89.1 | 87.8 | 82.2 | 83.3 | 60.4 | 69.2 | 95.3 | 37.7 | 51.2 |

Lastly, to gain deeper insights into the tasks that benefit most from our proposed D-Attn model, we present the detailed scores for MME (Fu et al., 2024) in Table 4, SEED (Li et al., 2023) in Table 5, and MMB (Liu et al., 2023) in Table 6. Our analysis reveals that our D-Attn model excels particularly in tasks requiring spatial and relational reasoning. Notable examples include (1) "position" in MME, (2) "Spatial Relation" in SEED, and (3) "object localization" and "spatial relationship" in MMB. In addition, our D-Attn model demonstrates strong performance on tasks involving OCR and document understanding. Specific examples include (1) "OCR" in MME, (2) "Text Understanding" in SEED, and (3) "ocr" and "structuralized image-text understanding" in MMB.

## 4 RELATED WORKS

Emerging large vision-language models (LVLMs) have made significant progress in visual understanding, particularly in Visual Question Answering (VQA). The predominant architectures can be summarized as different combinations of a vision encoder, an adapter, and a large language model (LLM). To name a few, LLaVA (Liu et al., 2024b), LLaVA NeXT (Liu et al., 2024a), LLaVA-OneVision (Li et al., 2024a), Instruct BLIP (Dai et al., 2023), BLIP3 (Xue et al., 2024), VILA (Lin et al., 2024), QWen2-VL (Wang et al., 2024), CuMo (Li et al., 2024b), Intern-XC-2.5 (Zhang et al., 2024), miniGemini (Li et al., 2024c), Cambrian-1 (Tong et al., 2024a), Phi-3 VL (Abdin et al.,

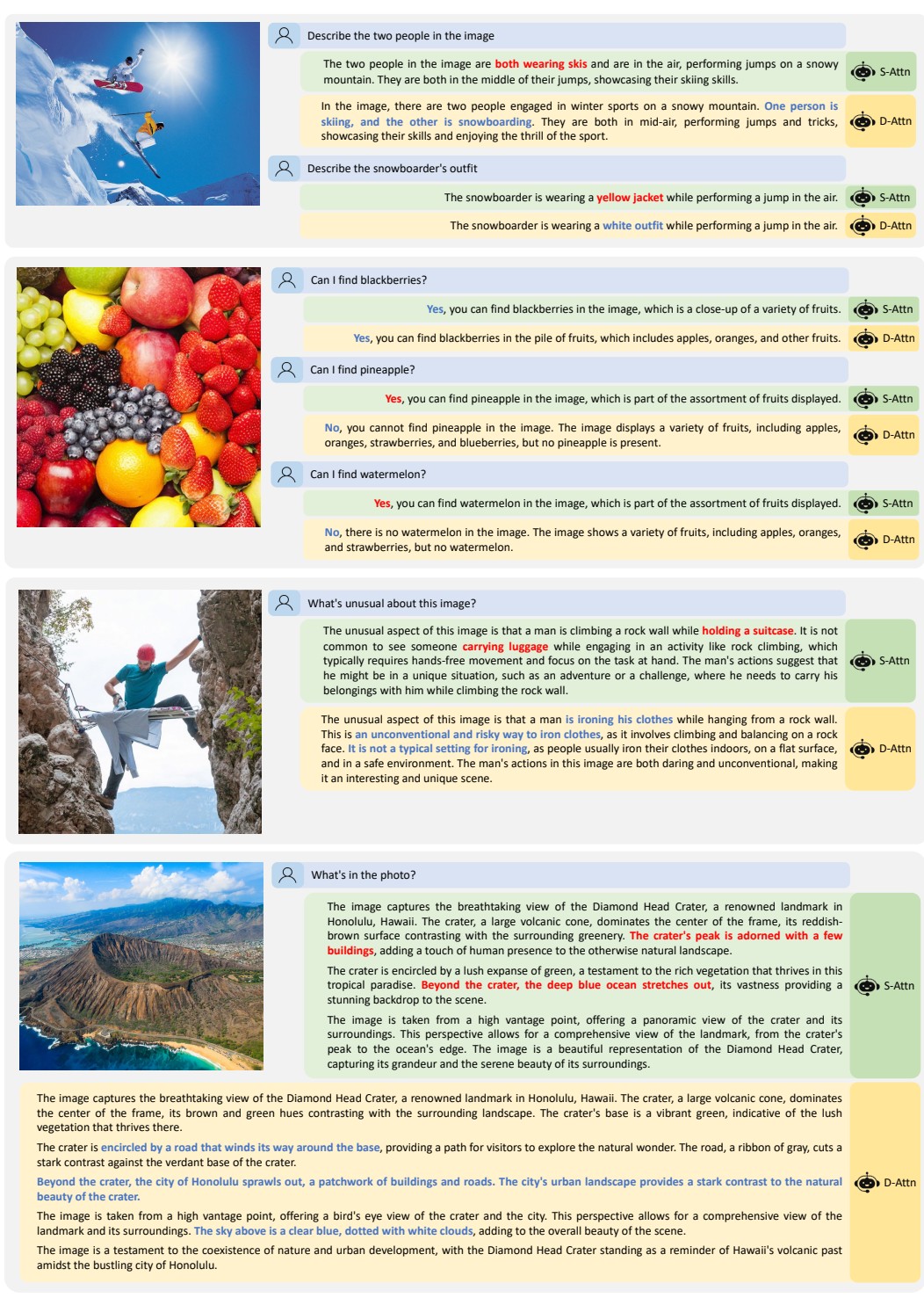

Figure 5: Qualitative comparisons between D-Attn and its Self-Attn (S-Attn) counterpart. Erroneous outputs from the S-Attn model are highlighted in red, while the accurate and preferred responses from D-Attn are highlighted in blue.

2024), Chameleon (Team, 2024), Molmo (Deitke et al., 2024), Phi-3.5-Vision (Abdin et al., 2024). Despite differences in data, vision encoders, or adapter, *all these works adhere to a decoder-only*

*LLM architecture that process visual and textual embeddings homogeneously* using the self-attention mechanism (Vaswani, 2017) within an LLM.

In contrast to predominant LVLM architectures, models like Flamingo (Alayrac et al., 2022; Awadalla et al., 2023), IDEFICS (Laurençon et al., 2024a), and LLaMA 3 (Dubey et al., 2024) integrate visual information into LVLMs via cross-attention mechanisms between textual and visual embeddings. These architectures share similarities with our proposed D-Attn, such as employing T2V Cross-Attention to incorporate visual data and achieving a computational complexity of $\mathcal{O}(|V|)$ for $|V|$ visual embeddings. However, this line of works differ notably in how they merge visual and textual modalities: by appending additional cross-attention modules or introducing tanh/sigmoiod gating to modulate visual information. These substantial architectural changes can compromise the integrity of the pre-trained LLM, potentially degrading its inherent capabilities. Indeed, Laurençon et al. (2024b) show in IDEFICS-2 that cross-attention architectures underperform decoder-only architectures, leading them to discard the cross-attention design. In this paper, we propose $\alpha$-weighting strategy equivalently derived from the native attention operations of LVLMs. $\alpha$-weighting introduces minimal architectural changes and requires no additional learnable parameters, ensuring the pre-trained LLM retains its full capability for competitive downstream performance.

## 5 CONCLUSION

In this paper, we introduced Decomposed Attention (D-Attn), a novel and general framework designed to process visual and textual embeddings differently within LVLMs. Through the decomposition of conventional causal self-attention in LVLMs, D-Attn reduces the computational complexity from $\mathcal{O}(|V|^2)$ to $\mathcal{O}(|V|)$ by diagonalizing V2V Self-Attn, and improve model performance by debiasing T2V Cross-Attn. To merge back visual and textual information, our proposed $\alpha$-weighting strategy preserves the capabilities of pre-trained LLMs with minimal modifications. Extensive experiments and rigorous analyses demonstrate that D-Attn consistently outperforms its S-Attn counterpart, offering both performance gains and substantial computational savings. Our contributions highlight the importance of handling visual and textual inputs differently, paving the way for more efficient and effective LVLMs.

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

## A  HYPER-PARAMETERS

In Table 7, we list key hyper-parameters for all three training stages and two LLMs, Mistral 0.3 7B and Gemma 2 9B. We use the same set of hyper-parameters for D-Attn models and their S-Attn counterparts. The weight decay and AdamW-related parameters are taken from LLaMA 2 (Touvron et al., 2023) technical report.

Table 7: Hyperparameters for three training stages and two types LLMs.

|  | **Stage 1** | **Stage 2** | | **Stage 3** | |
|---|---|---|---|---|---|
|  |  | Mistral 0.3 7B | Gemma 2 9B | Mistral 0.3 7B | Gemma 2 9B |
| lr adapter | 1e-3 | 5e-6 | 2e-5 | 5e-6 | 2e-5 |
| lr llm | 0.0 | 2e-6 | 1e-5 | 2e-6 | 1e-5 |
| lr vis-enc | 0.0 | 2e-7 | 1e-6 | 2e-7 | 1e-6 |
| weight decay | 0.0 | 0.1 | | 0.1 | |
| optimizer | AdamW | AdamW | | AdamW | |
| Adam $\beta_1$ | default (0.9) | 0.9 | | 0.9 | |
| Adam $\beta_2$ | default (0.999) | 0.95 | | 0.95 | |
| Adam $\epsilon$ | default (1e-8) | 1e-5 | | 1e-5 | |
| warmup ratio | 0.03 | 0.03 | | 0.03 | |
| lr scheduler | cosine | cosine | | cosine | |
| epochs | 1 | 1 | | 1 | |
| total batch size | 512 | 256 | | 128 | |
| dtype | bfloat16 | bfloat16 | | bfloat16 | |
| deepspeed | stage 2 | stage 3 | | stage 3 | |

