# OpenReview forum: "Rethinking Homogeneity of Vision and Text Tokens in Large Vision-and-Language Models"
_ICLR.cc/2025/Conference — Submitted to ICLR 2025_

### Official Review · Reviewer_KDE8 · 2024-11-01

**Soundness:** 2
**Presentation:** 3
**Contribution:** 2
**Rating:** 5
**Confidence:** 3

**Summary:**

This paper proposes the D-Attn framework for training LVLMs, with three key modifications to the standard architecture: diagonalized V2V attention, debiased positional encodings, and alpha-weighting. Experiments show superior performance relative to the standard framework under similar training settings.

**Strengths:**

The results show that the proposed framework boosts performance across a variety of tasks. The use of open models is appreciated. The visualizations are clear and benefit the paper’s exposition.

**Weaknesses:**

A number of points in the methodology raise questions:

Diagonalized V2V attention: As this is just applying FC layers to each visual input, calling this “attention” seems like a misnomer. I would expect this to impair performance on inputs containing multiple images which must be understood in context of each other – do the benchmarks tested contain multiple images interleaved with text? If they mostly contain single images, wouldn’t the proposed method be identical to standard attention?

Alpha-weighting: Section 2.4 shows that this is equivalent to standard LVLM attention, so what is being proposed? This seems to simply be another analytic expression for attention and not a novel component being proposed.

I find the demonstration in figure 5 unconvincing. The modality gap is a known phenomenon and may even be beneficial in some cases [1]. The fact that textual and visual embeddings cluster separately does not mean that a linear projection does not produce representations useful for downstream tasks ([1] is a counterexample of this).

[1] Liang et al. Mind the gap: Understanding the modality gap... NeurIPS 2022

**Questions:**

The paper claims diagonalized V2V attention is most advantageous e.g. with high-resolution input images – what about applying standard resizing to input images? Especially since this is a standard step for inputting images to CLIP-like visual encoders.

Debiased positional encoding: With the proposed method, what gives the LVLM information about the relative position of text and image? If there is none, doesn’t this mean the LVLM is blind to the position of an image within text?

---

> ### Author Response · Authors · 2024-11-22
>
> ### **1. V2V Diagonal-Attn for single image and multi-images**
> In the case of single image, V2V Diagonal-Attn is different from standard attention. Note that |V| represents the number of visual embeddings, not number of images as defined in L19, L74, L136, etc. For example, an image will be encoded into 576 visual embeddings by CLIP ViT-L/14-336. Standard Self-Attn in LVLM computes the attention between each pair of visual embeddings to model the contextual information. For our proposed V2V Diagonal-Attn, as described in Section 2.2, since the visual encoder has already modeled this contextual information, we propose not to re-learn it within the LLM, thereby saving substantial computations.
>
> In the case of multiple images, since V2V Self-Attn is for capturing the contextual information between visual embeddings, when diagonalized, we may expect weakened modeling of this contextual information. In practice, there is no noticeable performance degradation. Specifically, our model follows LLaVA's anyres strategy, where a high resolution image is divided into several patches, each encoded by CLIP independently. In this case, each patch has no such contextual information about other patches. However, as we can see in Table 2 of the paper, incorporating V2V Diagonal-Attn does not degrade performance.
>
> | Diag. Attn | GQA | VQA-T | MME | MMB | SEED-I | LLaVA-W | MMStar |
> |---|---|---|---|---|---|---|---|
> | N | 61.8 | 62.2 | 1533.1 | 70.3 | 70.5 | 70.7 | 36.8 |
> | Y | 63.4 | 63.4 | 1507.6 | 68.8 | 70.7 | 71.2 | 32.6 |
>
> To further confirm this observation, we train a VideoQA model following LLaMA-VID [[1]](https://arxiv.org/abs/2311.17043) , where the input is multiple frames (images) and the model has to model the contextual information across frames to answer the question. In this experiment, we again observed no performance degradation by incorporating V2V Diagonal-Attn.
>
> | Diag. Attn | ActQA | VChat CI | VChat DO | VChat CU | VChat TU | VChat CO |
> |---|---|---|---|---|---|---|
> | N | 47.0 / 2.3 | 2.2 | 2.2 | 2.7 | 2.1 | 2.3 |
> | Y | 46.8 / 2.3 | 2.4 | 2.2 | 2.8 | 2.0 | 2.6 |
>
> Our explanation of this phenomenon is that visual embeddings could exchange information indirectly via textual embeddings. Concretely, each textual embedding gathers visual information via T2V Cross-Attn at the L-th decoder layer. Therefore, at the (L+1)-th layer, visual embeddings indirectly exchange information via text embeddings in T2V Cross-Attn.
>
> ### **2. Computational advantage of V2V Diagonal-Attn for high-resolution images**
> V2V diagonal attention is most advantageous in the case of high |V| such as high resolution images. Although it is straightforward to reduce |V| by downsampling the input high-resolution images, it has been widely shown in previous works such as LLaVA [[1]](https://arxiv.org/abs/2310.03744) that a high-resolution input image is necessary for better performance. Simply downsampling images for reducing computation may degrade model performance. In contrast, our proposed V2V Diagonal-Attn reduces the overall computational complexity without compromising model performance as shown in Table 2 of the main paper.
>
> ### **3. Novelty and contribution of proposed $\alpha$-weighting**
> $\alpha$ weighting is a novel way of merging visual and textual information from T2V Cross-Attn and T2T Self-Attn, respectively. In cross-attention-based LVLMs, merging visual and textual information is an inevitable and critical challenge. As we point out in Section 2.4 and Section 4 of the paper, existing methods involve significant architectural changes or introduce additional parameters, which can break the integrity and degrade the performance of pre-trained LLMs. Retaining equivalence with conventional LVLM attention preserves the pretrained LLM’s capabilities and thus leads to superior performance. Compared with other existing merging strategies, alpha weighting achieves superior performance as shown in Table 3 of the paper.
>
> ### **4. Visual and textual modality gap**
> Thanks for pointing out the relevant work. Similar to referenced paper, we did not try to prove whether this modality gap is good or bad, nor propose closing this gap. Instead, this observation of modality gap shown in Figure 5 challenges the common belief in modern LVLM works such as LLaVA, where the authors claim that visual embeddings are projected into the same space as textual embeddings and thus could be processed homogenously within an LLM. This observation motivates us to process visual and textual embeddings differently within the LLM. We thereby propose the D-Attn framework to achieve this goal. We will revise the writing in the latest revision to make it more clear.

---

> ### Author Response · Authors · 2024-11-22
>
> ### **5. Relative position of text and image**
> In the standard LLaVA model, we have explored putting visual embeddings at different locations and observed no noticeable performance difference. Specifically, we try placing the input image before both system prompt and text query, after system prompt and before text query, as well as after both system prompt and text query. We do not observe substantial performance differences. This may imply that the position of an image within text does not play a crucial role.
>
> In the case where the relative position between visual and textual embeddings is absolutely necessary, it is easy to re-introduce positional info in our D-Attn framework while retaining the nice property of debiased positional embedding. Specifically, as shown in the figure below:
>
> - [Debiased positional encoding](https://imgur.com/FZtZ9Aj)
>
> we can set the position of all visual embeddings to a constant (0 here), and set the position of textual embeddings as usual (1, 2,...,5 here). In this setup, each textual embedding has unbiased positional weights toward all visual embeddings, while textual embeddings at different positions are aware of the positional difference with visual embeddings.
>
> ### **6. Naming of V2V Diagonal-Attn**
> Applying two $fc$ layers is the result of diagonalizing the standard V2V Self-Attn. This motivates us to reuse and share the $fc_v$ and $fc_o$ from the pre-trained attention module. We also inherit other designs in the self-attention module, such as pre-layernorm, post-layernorm, and skip connection. Overall, V2V Diagonal-Attn is more than just applying two fc layers. We therefore name this design V2V Diagonal-Attn as a hint of its origin and motivation.

---

> ### Comment · Reviewer_KDE8 · 2024-11-24
>
> Thank you for your detailed response. I will respond to the numbered points in turn below.
>
> 1, 2, 5.
> Thank you for clarifying the use of the “anyres” strategy, which is important for understanding your results. This should be explicitly mentioned and properly cited in the paper. At the moment the cited Liu et al. “Visual Instruction Tuning” is cited but not Liu et al “Improved Baselines with Visual Instruction Tuning” which discusses this strategy as LLaVa-1.5 (vs. LLaVa-1, which does not use it).
>
> Your observations regarding multiple visual embeddings (including multiple images) in context are interesting and will strengthen the paper.
> I am still confused by L144—150 which motivate V2V attention by saying that “each visual embedding already encapsulates contextual information from other visual embeddings”, which seems to contradict the fact that separate images (or separate patches using anyres) are encoded separately. Can you please clarify this?
>
> Regarding relative positioning of images and text, this is a useful clarification and I encourage the authors to add this to the revised paper.
>
> 3.
> I will appreciate further clarification here, as I think this does not answer my original question (also asked by reviewer tS4a). If alpha-weighting is mathematically equivalent to LVLM attention, what is being done differently (and what is being ablated in Table 3)?
>
> 4.
> Regarding the modality gap, I’m not sure I find this argument compelling.  LVLMs such as LLaVa apply a projection from visual tokens to be processed similarly to textual tokens, but this does not necessarily assume that a modality gap does not exist after the projection — just that they can then be processed with the same decoder for empirically satisfying results. Hence, the presence or degree of this gap does not seem obviously relevant.
>
> 6.
> While I understand the motivation behind the name “V2V Diagonal-Attn”, the use of fully-connected layers, layer norm, and residual connections were already standard in pre-transformer architectures, and normally the use of the term "attention" implies the use of the attention mechanism built from keys, queries, and values. Hence I still find this name somewhat misleading.

---

> > ### Author Response · Authors · 2024-11-25
> >
> > Thanks for your prompt reply and valuable feedback. We will revise the writing to (1) mention that we use the anyres strategy for a high resolution image and cite LLaVA 1.5, (2) describe the observations regarding multiple visual embeddings (including multiple images) in context, and (3) include the discussion regarding relative positioning of images and text. For the remaining questions, we reply to each one in more detail as follows.
> >
> > ### **1. Clarification of the claim "Each visual embedding already encapsulates contextual information from other visual embeddings"**
> > In the paper, we develop the idea around the single-image scenario. Given a single image, each visual embedding already encapsulates contextual information from other visual embeddings, as these visual embeddings are created by passing the single image through a pre-trained encoder. We agree with the reviewer that this claim may not hold for multiple images. In this scenario, however, visual embeddings from different images could still exchange information indirectly via textual embeddings as explained in the previous reply. We will revise the paper to clarify this.
> >
> > ### **2. Explanation of $\alpha$ weighting and Table 3**
> > For architectures that handle visual and textual tokens with two separate T2T and T2V attentions, e.g., Flamingo, LLaMA 3, or our proposed D-Attn, merging information from these two attentions is an inevitable and critical challenge. In Table 3, we compare with different merging strategies and found that the merging strategies in existing literature (e.g., flamingo) often hurt performance. The key to superior performance is maintaining the mathematical equivalence to LLM's self-attention, which is novel and non-trivial in the field of LVLMs. To the best of our knowledge, this is the first work to mathematically derive alpha-weighting for this class of architecture, and experimentally (Table 3) demonstrate its advantage over existing merging strategies.
> >
> > ### **3. Modality gap**
> > We will tone down about the discussion regarding the modality gap in current LVLMs. The "mind the Gap" paper and Figure 5 confirm the existence of modality gap. This observation of modality gap motivates us to re-think about the inherent difference between visual and textual embeddings, and thus the proposal of our D-Attn to process them differently. We will emphasize the "process differently" part in the latest revision.
> >
> > ### **4. Naming of V2V Diagonal-Attn**
> > Thanks for pointing this out, we did not realize the term "diagonal-attn" could be confusing in this way. We will think carefully about what could be a better name for this part. We agree with the reviewer that the use of fully-connected layers, layer norm, and residual connections were already standard in pre-transformer architectures. For removing the term "attention", one concern we have is that current "Diagonal-Attn" is easy and intuitive to implement with minimal implementation changes following the huggingface transformers package. On the other hand, describing it as a combination of fc layers, normalization, skip connection, etc, can cause another confusion about the implementation details, e.g., the hyperparameters and architecture. It may be harder for readers to mirror the current "Diagonal-Attn" with the original attention block implementation. We thank the reviewer for raising this concern, and will re-think what's a better name to not cause confusion.

---

> > > ### Comment · Reviewer_KDE8 · 2024-11-26
> > >
> > > Thank you for your thoughtful response. I have two follow-up questions on points 1 and 2:
> > >
> > > 1. To be clear, by single-image you mean a single low-resolution image, correct? Because if I understand correctly, with a higher-resolution image, the anyres strategy would encode it as multiple separate embeddings which do not "already encapsulate information from other visual embeddings". Therefore it seems like this statement should be further qualified.
> > >
> > > 2. Am I correct that alpha-weighting thus refers simply to directly using visual tokens in the self-attention mechanism? If so, this seems to be equivalent to LLaVa's approach to directly insert projected visual embeddings into a language model (which they explicitly state is an alternative to the gated merging strategies [1]). Is this different, and if not, why is it being proposed as a novel contribution?
> > >
> > > [1] Liu et al. 2024 "Visual Instruction Tuning" Sec 4.1: "More sophisticated schemes to connect the image and language representations can also be considered, such as gated cross-attention in Flamingo and Q-former in BLIP-2."

---

> ### Author Response · Authors · 2024-11-27
>
> Thanks for the prompt reply and detailed comments, we address the follow-up questions below.
>
> ### **1. Clarification of the term "single-image" used in the paper**
> Yes, that is correct. Here "single-image" refers to a single low-resolution image that is fed into CLIP as a whole. A single high-resolution image divided into patches following the anyres strategy should be considered as "multiple-images" because each divided patch is encoded independently with CLIP.
>
> We will clarify this and clearly define what "single-image" means in the paper. We will also include the additional experimental results for multiple images in previous response to further justify that our D-Attn model also works well in multi-image scenarios.
>
> ### **2. Difference between proposed D-Attn and conventional S-Attn in LLaVA**
> No, that is incorrect. The proposed D-attn is different from LLaVA's self-attention mechanism:
> 1. LLaVA applied one self-attention to the concatenated visual and text tokens, while D-attn decompose this architecture and applied three attention operations (V2V, V2T, T2T) with a merging strategy,
> 1. we replace the V2V self-attention with Diagonal-Attn/FCs so that the computational complexity is reduced from $\mathcal{O}(|V|^{2})$ to $\mathcal{O}(|V|)$.
> 1. We replaced the original position encoding with the proposed debiased positional encoding which boosts model performance.
>
> D-attn would degenerate to LLaVA's self-attention **only if 2) and 3) are both NOT applied**. Given that 1), 2), 3) are all applied, the proposed D-attn is significantly different from LLaVA's attention mechanism. Although LLaVA claimed alternative designs of incorporating visual information, no experiments were presented to verify the effectiveness of these designs. In fact, when comparing the $\alpha$-weighting strategy to existing merging strategies for LVLM with multiple attention operations (e.g., Flamingo), the proposed merging strategy is novel and it performs better than other merging strategies as shown in Table 3.
>
> We really appreciate your feedback and your patience for the discussion. If there is further question about this contribution or the difference between D-Attn with other methods like S-Attn, Flamingo, LLaVA, etc, we are happy to provide further clarification and context.

---

> > ### Comment · Reviewer_KDE8 · 2024-11-27
> >
> > Thank you for your response.
> > To be clear, in question 2 I was not referring to the proposed "Diagonal-Attn" or debiased positional encodings, but only to the alpha-weighting strategy, as I agree that the former as clearly different from LLaVa's strategy. My question was not about the overall difference between D-Attn and LLaVa, but only the novelty of the alpha-weighting component, which is still not clear to me.
> >
> > I believe the discussion provides useful context for the paper, and as some of my concerns remain with the paper as-is (particularly the alpha-weighting contribution and modality gap relevance) I will keep my score.

---

> > > ### Author Response · Authors · 2024-11-27
> > >
> > > ### **1. Novelty of alpha weighting**
> > > Thanks for your feedback and the clarification of your question. We try our best to explain the novelty of alpha-weighting as follows:
> > >
> > > 1. To process visual embeddings differently in LVLM and gain the computation and performance benefits (e.g. our V2V Diagonal-Attn and debiased positional encoding), we have to decompose self-attention and compute T2T Self-Attn and T2V Cross-Attn separately.
> > > 2. Due to the decomposition, we need a good strategy to merge the decomposed T2T Self-Attn and T2V Cross-Attn.
> > > 3. We find existing merging strategies (e.g. Flamingo) lead to suboptimal performance as shown in Table 3 of the paper.
> > > 4. We observe that existing merging strategies involve significant changes, which can break the integrity and degrade the performance of pre-trained LLMs.
> > > 5. Based on the above observation, we derive and propose $\alpha$-weighting, an explicit merging strategy mathematically equivalent to the implicit merging operations within LLM's self-attention.
> > > 6. We implement and empirically validate the effectiveness of $\alpha$-weighting in Table 3.
> > >
> > > To the best of our knowledge, we consider it novel for the following reasons:
> > > 1. **Novel idea**: We are the **first** to propose the idea that maintaining this equivalence for merging is the key to better performance.
> > > 2. **Non-trivial derivation**: We are the **first** to mathematically derive this merging strategy, which we refer to as $\alpha$-weighting throughout the paper and our discussion.
> > > 3. **Easy implementation**: $\alpha$-weighting is in a simple form, which is easy to implement and compute.
> > > 4. **Better performance**: $\alpha$-weighting performs better than existing merging strategies.
> > >
> > > Please let us know if you have any further concerns about $\alpha$-weighting merging strategy. We are happy to provide more clarification or context about this.
> > >
> > > ### **2. More Context**
> > > To help understand the novelty of $\alpha$-weighting merging strategy, we provide more context about major LVLM architectures as follows:
> > >
> > > There exist two ways of incorporating visual information in modern LVLMs: (1) self-attention for concatenated visual and textual embedding, and (2) Separate T2T Self-Attn for textual embeddings and T2V Cross-Attn for visual embeddings.
> > > 1. Self-attention only: Most modern LVLMs fall into this category, such as LLaVA, InternVL, QwenVL, etc. This line of work applies only one self-attention on concatenated visual and textual tokens without any explicit merging operation. Therefore, there is NO need to consider the strategy for merging T2T Self-Attn and T2V Cross-Attn. This line of work lacks the flexibility to process visual and textual embeddings differently. The lack of flexibility hinders the opportunity to process visual embeddings differently, such as V2V Diagonal-Attn for reducing computation, and debiased positional encoding for enhancing model performance.
> > > 2. Separate T2T Self-Attn & T2V Cross-Attn: Only a few LVLMs fall into this category, such as Flamingo, LLaMA 3, and our proposed D-Attn. This line of work explicitly separates T2T Self-Attn and T2V Cross-Attn. Therefore, it is inevitable and critical to consider the strategy for merging T2T Self-Attn and T2V Cross-Attn. This line of work provides the flexibility to process visual and textual embeddings differently.
> > >
> > > In this paper, in order to process visual and textual embeddings differently, we adopt approach (2) with separate T2T Self-Attn & T2V Cross-Attn. We found in Table 3 of the paper that the merging strategies in existing literatures (e.g., flamingo) often hurt performance. The key to superior performance is maintaining the mathematical equivalence to LLM's self-attention ($\alpha$-weighting), which is novel and non-trivial in the field of LVLMs. To the best of our knowledge, this is the first work to mathematically derive $\alpha$-weighting for this class of architecture, and experimentally (Table 3) demonstrate its advantage over existing merging strategies.

---

### Official Review · Reviewer_1sRA · 2024-11-03

**Soundness:** 3
**Presentation:** 3
**Contribution:** 3
**Rating:** 6
**Confidence:** 5

**Summary:**

This paper explores the network architecture of LVLMs regarding visual and textual embeddings. The authors propose a Decomposed Attention mechanism that treats visual and textual embeddings differently to achieve faster training and better performance. An adaptive α-weighting method is introduced to fuse visual and textual information. Experiments on popular benchmarks demonstrate the effectiveness of the proposed methods.

**Strengths:**

- The problem addressed in this work is both interesting and fundamental.
- The paper is well-illustrated and easy to understand.
- The proposed Decomposed Attention and α-weighting are simple yet effective.

**Weaknesses:**

- Limited novelty of Decomposed Attention: Using cross-attention to fuse visual and textual embeddings is not novel, as it has been explored in previous VLMs, such as Flamingo [1]. Although this work investigates cross-attention in popular LVLM architectures, like LLaVA [2], the technical contribution appears limited.
- Lack of analysis on α-weighting: Another major contribution is the α-weighting mechanism for fusing visual and textual information, with ablation studies demonstrating its effectiveness compared to Tanh and Sigmoid. However, there is no in-depth analysis of α-weighting.
- Unclear motivation: In Figure 5, the authors suggest that the primary motivation is that textual and visual embeddings occupy different regions in the feature space of LLaVA1.6, which is already revealed in Mind the Gap[3] and well-known in this field. However, a similar feature space analysis is not presented in the proposed model with decomposed attention, which leads to uncertainty about whether the improvements are driven by other factors or the claimed ones.

Reference:

[1]Flamingo: a Visual Language Model for Few-Shot Learning.

[2]Improved Baselines with Visual Instruction Tuning.

[3]Mind the Gap: Understanding the Modality Gap in Multi-modal Contrastive Representation Learning.

**Questions:**

- For the first weakness, could the authors clarify the main contribution of this work?
- Regarding the second weakness, could the authors provide a deeper analysis of the underlying mechanism of the α-weighting strategy? For instance, reporting the statistics of αV across various benchmarks such as VQA-T, SQA-I, and GQA could shed light on how the DA model prioritizes visual and textual features.
- For the last weakness, could authors provide more evidence that the disentangled visual and textual feature learning benefits the model via proposed methods?

---

> ### Author Response · Authors · 2024-11-22
>
> ### **1. The main contribution and novelty of this paper**
> $\alpha$ weighting is a novel way of **merging** visual and textual information from T2V Cross-Attn and T2T Self-Attn, respectively. We agree with the reviewer that using cross-attention between visual and textual embeddings has been explored in, e.g. Flamingo, and did not claim these parts as novel or our contribution in the main paper. We discussed the nuance difference between our D-Attn framework and other cross-attention-based LVLMs in L486-499. Compared with other existing merging strategies, $\alpha$ weighting achieves superior performance as shown in Table 3 of the paper.
>
> In addition to the novel $\alpha$-weighting strategy, in this paper we propose the novel V2V Diagonal-Attn and debiased positional encoding. V2V Diagonal-Attn substantially reduces computation from $\mathcal{O}(|V|^2)$ to $\mathcal{O}(|V|)$ without compromising performance. Debiased positional encoding removes undesirable positional bias between textual and visual embeddings, leading to consistent performance improvements. Ablations for these two major contributions are presented in Table 2 of the paper.
>
> ### **2. Motivation of decomposed attention**
> Thanks for pointing out the relevant work. Similar to referenced paper, we did not try to prove whether this modality gap is good or bad, nor propose closing this gap. Instead, this observation of modality gap shown in Figure 5 challenges the common belief in modern LVLM works such as LLaVA, where the authors claim that visual embeddings are projected into the same space as textual embeddings and thus could be processed homogenously within an LLM. This observation motivates us to process visual and textual embeddings differently within the LLM. We thereby propose the D-Attn framework to achieve this goal. We will revise the writing in the latest revision to make it more clear.
>
> The improvement is driven by our novel designs, including V2V Diagonal-Attn, debiased positional encoding, and alpha weighting, to process visual and textual embeddings differently. The decomposition per se does not directly bring performance improvement. Rather, it gives us more control and flexibility, thus enabling us to derive and incorporate the novel designs mentioned above for performance improvement.
>
> ### **3. Analysis of alpha-weighting**
>
> Thanks for the suggestion. We show $\alpha_V$ on these benchmarks across attention heads and LLM layers in the following links:
> - [SQA-I](https://imgur.com/lR3txQd)
> - [GQA](https://imgur.com/Gr9hySs)
> - [VQA-T](https://imgur.com/95715QT)
>
> For each layer, we sort $\alpha_V$ across heads for better visualization. We can see that GQA and VQA-T both have high $\alpha_V$ across heads and layers, while SQA-I has much lower $\alpha_V$ . This observation is in consensus with MM-Star [[1]](https://arxiv.org/abs/2403.20330), where the authors point out that many questions in SQA do not require visual information to answer. We will add these figures to the paper in the latest revision.

---

> > ### Comment · Reviewer_1sRA · 2024-11-23
> >
> > Thank you for your detailed response. I appreciate the effort you have put into addressing the concerns raised. However, I only have one question about the Analysis of alpha-weighting. Based on the provided visualization,
> >
> > Could authors provide a similar visualization with the cascade variant in the main paper Table3? I think a comparison between D-Attn and the cascade variant can support the authors' claims.

---

> ### Author Response · Authors · 2024-11-25
>
> Thanks for your prompt reply and clarification. We explain more about the merging strategies as follow.
>
> In terms of the cascade variant, the concept of merging weight between T2T Self-Attn and T2V Cross-Attn is not applicable. In the cascade variant (described in L412-413), input textual embeddings first perform T2T Self-Attn, and then the **output** of the T2T Self-Attn performs T2V Cross-Attn. These two operations were performed in series and there is no such weighting as our alpha-weighting strategy in D-Attn or as standard Self-Attn.
>
> Nevertheless, we analyze the sigmoid variant, where T2V Cross-Attn and T2T Self-Attn are weighted summed with learnable gates $\sigma$ as described in L414-415. Different from our $\alpha$-weighting, the merging weights of $\sigma$ gate are fixed after training and are independent of data. Therefore, it is unable to adaptively adjust the weighting between visual and textual embedding depending on the task. Furthermore, in our latest analysis, the learned sigmoid gates have a mean and standard deviation of 0.620 and 0.00255, respectively, across heads and layers, unlike our alpha weighting strategy that has greater expressibility across heads, layers, and datasets as shown in our previous reply.

---

> > ### Comment · Reviewer_1sRA · 2024-11-26
> >
> > Thanks the authors for their response. After reading the responses to all reviewers, I think the motivation for D-Attn is not convincing enough. However, the proposed method is simple and effective, which could potentially serve as a fundamental architecture for further MLLMs. For the above reasons, I think my current rating is appropriate and will raise my confidence.

---

### Official Review · Reviewer_tS4a · 2024-11-03

**Soundness:** 3
**Presentation:** 2
**Contribution:** 3
**Rating:** 6
**Confidence:** 4

**Summary:**

The authors propose Decomposed Attention, a novel approach that treats visual and textual information differently within Large Vision Language Models (LVLMs). By eliminating attention between different tokens related to visual data, they enhance efficiency, demonstrating significant performance improvements across various multimodal tasks.

**Strengths:**

1) The authors clearly articulate the rationale behind their design choices. They explain the motivation for introducing diagonal attention and the different positional embedding methods for visual and textual embeddings. This depth of reasoning strengthens the justification for their approach.

2) The proposed method is highly effective in terms of efficiency, which the authors substantiate both conceptually and empirically. They discuss the reduction in computational complexity achieved through the use of diagonal attention and illustrate improvements in metrics such as $max|V|$ and sec/it, as shown in Table 2.

**Weaknesses:**

1) The meaning of alpha-weighting is unclear. From the perspective of text tokens, it does not appear to differ from the existing operations. It is ambiguous why the authors refer to this as alpha-weighting and describe it as a distinct approach.

2) There is insufficient analysis of the proposed method's effectiveness. It seems that the use of debiased positional encodings significantly contributed to the performance improvement in image benchmarks, yet this is not adequately mentioned or discussed. Additionally, the varying extent of performance improvement across different image benchmarks is not addressed, leaving the discussion lacking regarding which tasks the proposed method significantly impacts.

3) The analysis of the benefits gained from treating vision embeddings and text embeddings differently is inadequate. The authors only mention performance improvements in image benchmarks, without providing insight into how the proposed method influences task execution. They claim that there is a misalignment between modalities in the embedding space of LLaVA-1.6, but do not discuss whether their method addresses this issue, making the relevance of this analysis questionable.

**Questions:**

1) In which specific tasks does the proposed method demonstrate improvements? Beyond the numerical performance improvements shown in Table 1, a discussion on why it is particularly effective for certain tasks is needed.

2) The authors present an analysis of the misalignment between vision and text tokens alongside Figure 5, stating, "this discrepancy cannot be easily mitigated by a simple adapter layer applied to the visual embeddings." Therefore, does the authors’ proposed method mitigate this discrepancy?

3) Line 221 states, "These approaches involve significant architectural changes..." Why is this considered a disadvantage? If the vision encoder is to be added for further training on multimodal data, then an architectural change seems acceptable.

4) While the authors describe alpha-weighting differently, the operation of text tokens attending to vision and text tokens remains the same as before. Is it valid to claim that this is a new proposal?

5) The information provided in Figure 4 duplicates what can be obtained from Table 1. Since Figure 4 only adds a visual effect without providing additional information, it may not be necessary to include it in the paper.

6) In Table 2, the term "debiased positional encodings" is used, while Line 409 refers to them as "biased positional encodings." Additionally, Section 2 uses the term "learnable positional encodings" without specifying either. The lack of consistency in terminology may confuse readers.

---

> ### Author Response · Authors · 2024-11-22
>
> ### **1. Novelty and contribution of proposed $\alpha$-weighting merging strategy**
> $\alpha$ weighting is a novel way of merging visual and textual information from T2V Cross-Attn and T2T Self-Attn, respectively. We agree with the reviewer that the operations of T2V Cross-Attn and T2T Self-Attn remain the same and did not claim these parts as novel or our contribution in the main paper. Compared with other existing merging strategies, $\alpha$ weighting achieves superior performance as shown in Table 3 of the paper.
>
> ### **2. Disadvantage of other merging strategies**
> Significant architectural changes are disadvantages in LVLM as these changes can compromise the integrity of the pre-trained LLM, potentially degrading its inherent capabilities. As we show in Table 3 of the paper, other merging strategies involving more architectural changes lead to suboptimal performance compared to our $\alpha$-weight strategy, which introduces minimal architectural/operational changes and retains equivalence with the native LVLM attention. Thanks for pointing this out, and we will revise the writing to make it more clear.
>
> ### **3. Analysis of the proposed method's effectiveness**
> We present the broken-down scores from MME, SEED, and MMB in the tables below.
>
> | MME | existence | count | position | color | posters | celebrity | scene | landmark | artwork | OCR |
> |---|---|---|---|---|---|---|---|---|---|---|
> | S-Attn | 190.0 | 165.0 | 121.7 | 180.0 | 134.4 | 161.2 | 166.3 | 157.8 | 128.0 | 102.5 |
> | D-Attn | 195.0 | 170.0 | 143.3 | 195.0 | 161.6 | 172.6 | 163.0 | 166.8 | 137.0 | 132.5 |
>
> | SEED | Scene Understanding | Instance Identity | Instance Location | Instance Attributes | Instances Counting | Spatial Relation | Instance Interaction | Visual Reasoning | Text Understanding |
> |---|---|---|---|---|---|---|---|---|---|
> | S-Attn | 76.9 | 74.5 | 74.7 | 67.3 | 64.2 | 57.8 | 73.2 | 76.1 | 44.7 |
> | D-Attn | 78.1 | 78.2 | 77.5 | 68.6 | 67.5 | 61.0 | 73.2 | 80.9 | 65.8 |
>
> | MMB | action recognition | attribute comparison | attribute recognition | celebrity recognition | function reasoning | future prediction | nature relation | object localization | ocr | social relation | spatial relationship | structuralized image-text understanding |
> |---|---|---|---|---|---|---|---|---|---|---|---|---|
> | S-Attn | 88.8 | 70.4 | 83.7 | 78.7 | 74.6 | 50.0 | 70.8 | 50.6 | 66.6 | 83.7 | 28.8 | 33.3 |
> | D-Attn | 90.7 | 77.2 | 89.1 | 87.8 | 82.2 | 50.0 | 83.3 | 60.4 | 69.2 | 95.3 | 37.7 | 51.2 |
>
> Overall, we found that D-Attn model is particularly strong on tasks involving spatial and relational reasoning, such as (1) "position" in MME, (2) "Spatial Relation" in SEED, and (3) "object localization" and "spatial relationship" in MMB. It also performs well on tasks involving OCR and document understanding, such as (1) "OCR" in MME, (2) "Text Understanding" in SEED, and (3) "ocr" and "structuralized image-text understanding" in MMB.
>
> ### **4. Modalities misalignment**
> In this paper, we did not try to prove whether this modality gap is good or bad, nor propose closing this gap. Instead, this observation of modality gap shown in Figure 5 challenges the common belief in modern LVLM works such as LLaVA, where the authors claim that visual embeddings are projected into the same space as textual embeddings and thus could be processed homogenously within an LLM. This observation motivates us to process visual and textual embeddings differently within the LLM. We thereby propose the D-Attn framework to achieve this goal. We will revise the writing in the latest revision to make it more clear.
>
> ### **5. Writing and presentation issues**
> The purpose of Figure 4 is to highlight the performance gain of proposed D-Attn models against their S-Attn counterparts, while Table 1 is for the comparison with other SoTA methods with detailed numbers across a wide range of image benchmarks. We consider revising the presentation or replacing Figure 4 with other analyses in the latest revision. For other writing issues, we greatly appreciate the reviewer's efforts in reading the paper in detail. We will fix the inconsistencies in the latest revision to make the presentation more clear.

---

> > ### Author Response · Authors · 2024-11-25
> >
> > Thank you for your valuable feedback on our submission. We have provided a detailed response to your comments and would appreciate any further clarifications or thoughts you may have. As the discussion period is ending soon, if there is anything else you would like us to address further, please let us know. We look forward to your input and greatly value your feedback.

---

> > > ### Comment · Reviewer_tS4a · 2024-11-26
> > >
> > > Thank you for the detailed response to my initial review. Some of my concerns have been addressed, but the weaknesses I raised earlier have not all been fully resolved.
> > >
> > > **Weakness 1**
> > >
> > > The authors mentioned in their response:
> > > >"... did not claim these parts as novel or our contribution in the main paper."
> > >
> > > However, in the paper’s abstract, they stated:
> > > > "we introduce an α-weighting strategy to merge visual and textual information ..."
> > >
> > > If this is not meant to be considered a contribution, shouldn’t it be clarified as such in both the abstract and introduction? As it currently stands, it seems to be presented as part of the paper’s contribution.
> > >
> > > **Weakness 3**
> > >
> > > The analysis provided is excellent, and including it in the paper would effectively highlight the strengths of the proposed method. However, it does not seem to have been reflected in the paper yet.
> > >
> > > **Question 2**
> > >
> > > If the section only serves to show motivation, I think it is not really necessary to place it here. It doesn’t seem to inspire the readers or demonstrate the effectiveness of the method proposed in the paper.

---

> ### Author Response · Authors · 2024-11-27
>
> Thanks for the prompt reply and detailed comments, we address the follow-up questions below.
>
> ### **1. Contribution of $\alpha$-weighting merging strategy**
> In our previous response, the complete sentence of the statement is:
> > _"We agree with the reviewer that the operations of T2V Cross-Attn and T2T Self-Attn remain the same and did not claim these parts as novel or our contribution in the main paper."_
>
> In the sentence, **"these parts"** refers to **the operation of T2V Cross-Attn** and the **the operation of T2T Self-Attn**. This statement does not contradict our claim of contribution in the paper regarding alpha-weighting, which is **a merging strategy for merging the outputs from T2V Cross-Attn and T2T Self-Attn**.
>
> If there is any question about this contribution, we are happy to provide further clarification and context.
>
> ### **2. Figure 5 and the analyses of detailed scores from MME, SEED, and MMB**
>
> Thanks for the suggestions. We have removed Figure 5, and replaced it with the analyses of detailed scores from MME, SEED, and MMB in the manuscript.
>
> ### **3. More Context**
> To help understand the novelty and contribution of $\alpha$-weighting merging strategy, we provide more context about major LVLM architectures as follows:
>
> There exist two ways of incorporating visual information in modern LVLMs: (1) self-attention for concatenated visual and textual embedding, and (2) Separate T2T Self-Attn for textual embeddings and T2V Cross-Attn for visual embeddings.
> 1. Self-attention only: Most modern LVLMs fall into this category, such as LLaVA, InternVL, QwenVL, etc. This line of work applies only one self-attention on concatenated visual and textual tokens without any explicit merging operation. Therefore, there is NO need to consider the strategy for merging T2T Self-Attn and T2V Cross-Attn. This line of work lacks the flexibility to process visual and textual embeddings differently. The lack of flexibility hinders the opportunity to process visual embeddings differently, such as V2V Diagonal-Attn for reducing computation, and debiased positional encoding for enhancing model performance.
> 2. Separate T2T Self-Attn & T2V Cross-Attn: Only a few LVLMs fall into this category, such as Flamingo, LLaMA 3, and our proposed D-Attn. This line of work explicitly separates T2T Self-Attn and T2V Cross-Attn. Therefore, it is inevitable and critical to consider the strategy for merging T2T Self-Attn and T2V Cross-Attn. This line of work provides the flexibility to process visual and textual embeddings differently.
>
> In this paper, in order to process visual and textual embeddings differently, we adopt approach (2) with separate T2T Self-Attn & T2V Cross-Attn. We found in Table 3 of the paper that the merging strategies in existing literatures (e.g., flamingo) often hurt performance. The key to superior performance is maintaining the mathematical equivalence to LLM's self-attention ($\alpha$-weighting), which is novel and non-trivial in the field of LVLMs. To the best of our knowledge, this is the first work to mathematically derive $\alpha$-weighting for this class of architecture, and experimentally (Table 3) demonstrate its advantage over existing merging strategies.

---

> ### Comment · Reviewer_tS4a · 2024-12-03
>
> Thank you for the detailed explanation and revisions made to the paper. I partially agree with the authors regarding their contributions. I was impressed by their active incorporation of the review comments into the paper. I have raised the score by 1 point.

---

### Official Review · Reviewer_tnid · 2024-11-04

**Soundness:** 3
**Presentation:** 3
**Contribution:** 2
**Rating:** 6
**Confidence:** 3

**Summary:**

This paper introduces Decomposed Attention (D-Attn), a novel framework for processing visual and textual embeddings differently within Large Vision-Language Models (LVLMs). The key insight is that visual and textual inputs are inherently different and should be treated accordingly. The paper proposes three main technical contributions: (1) diagonalization of visual-to-visual self-attention to reduce computational complexity from O(|V|²) to O(|V|), (2) debiasing of positional encodings in text-to-visual cross-attention, and (3) an α-weighting strategy for merging visual and textual information. The authors demonstrate improved performance across multiple benchmarks while significantly reducing computational costs.

**Strengths:**

1. **Novel Insight and Problem Formulation**
   - Clearly identifies and addresses a fundamental assumption in current LVLMs about homogeneous processing of visual and textual tokens
   - Provides compelling arguments for why visual and textual embeddings should be treated differently

2. **Technical Innovation**
   - Introduces an elegant solution for reducing computational complexity through diagonal V2V attention
   - Proposes a mathematically sound α-weighting strategy that maintains model capabilities
   - Successfully addresses positional encoding bias issues

3. **Empirical Validation**
   - Comprehensive experimental evaluation across multiple benchmarks
   - Thorough ablation studies demonstrating the contribution of each component
   - Significant computational efficiency gains while maintaining or improving performance

4. **Reproducibility**
   - Clear implementation details and hyperparameters provided
   - Uses publicly available datasets and models
   - Code and models promised to be released

**Weaknesses:**

1. **Limited Directional Flexibility**
   - The current formulation assumes a fixed direction of information flow (text as query, vision as key/value)
   - Does not adequately address scenarios where vision needs to query text or other vision tokens
   - May limit the model's capability in vision-centric tasks (e.g., visual search, face matching)

2. **Theoretical Foundations**
   - The justification for diagonal V2V attention could be strengthened with more theoretical analysis
   - Limited discussion on the potential information loss from diagonal approximation
   - The α-weighting derivation could benefit from more detailed mathematical exposition

3. **Input Order Dependency**
   - Current architecture may not handle cases where text follows vision in the input sequence
   - Lacks explicit discussion on maintaining causality while allowing flexible token ordering
   - Need for more experimental validation with varied input orderings

**Questions:**

1. How would the proposed D-Attn framework handle cases where visual tokens need to attend to text tokens (V2T attention)? Could the framework be extended to support bi-directional cross-modal attention while maintaining its computational efficiency?

2. The paper assumes text tokens as queries and visual tokens as keys/values. How would this affect tasks where visual information needs to act as queries (e.g., visual search, image matching)? Could you elaborate on potential extensions to support such scenarios?

3. In cases where text tokens appear before visual tokens in the input sequence, how does the current architecture maintain causal attention while allowing meaningful cross-modal interactions? Could you provide experimental results for such cases?

4. Have you considered a more general framework where the roles of query/key/value are dynamically assigned based on the task or context? What would be the computational implications of such an approach?

5. Could you provide more theoretical justification for the diagonal approximation in V2V attention? Specifically, what types of visual relationships might be lost, and in what scenarios might this approximation be suboptimal?

---

> ### Author Response · Authors · 2024-11-23
>
> ### **1. Textual embeddings placed before visual embeddings**
> In modern LVLMs such as LLaVA, textual embeddings are always placed after visual embeddings in both training [[1]](https://github.com/haotian-liu/LLaVA/blob/main/llava/train/train.py#L320) and inference [[2]](https://github.com/haotian-liu/LLaVA/blob/main/llava/eval/model_vqa_loader.py#L46). Due to the causal attention in LLM, visual embeddings will not act as query or attend to the textual embeddings appended afterward. Following this design, we derive our D-Attn framework assuming visual embeddings placed in front of textual embeddings as shown in Figure 1 of the paper.
>
> If placing textual embeddings in front of visual embeddings is absolutely necessary, our D-Attn framework is flexible enough to deal with this scenario while maintaining its performance and computational advantages. Consider the scenario where visual embeddings are inserted between two segments of textual embeddings, the causal attention is illustrated in the figure below:
>
> - [Placing text before image](https://imgur.com/9FCBULN)
>
> Following our D-Attn framework, we can merge the information from blocks 2 and 3, as well as blocks 4, 5, and 6 via our proposed alpha-weighting strategy. Similarly, V2V-Attn in block 3 can be diagonalized to save computation. This is a straightforward extension of our D-Attn framework, which maintains causal attention property.
>
> We have explored the design of placing textual embeddings before visual embeddings in our early stage of development and found no noticeable performance gain. We therefore decide to place the visual embeddings at the beginning following LLaVA's design.
>
> | Visual embeddings | SQA-I | MME | GQA | POPE | VQA-T | SEED | MMStar |
> |---|---|---|---|---|---|---|---|
> | Beginning | 68.57 | 1443.95 | 61.13 | 86.31 | 55.45 | 65.36 | 33.66 |
> | Middle | 68.72 | 1447.91 | 60.84 | 86.05 | 54.05 | 65.78 | 33.60 |
>
> ### **2. Bi-directional cross-modal attention**
>
> To the best of our knowledge, modern LVLMs (models developed after LLaVA) typically do not adopt bi-directional attention design. It is likely because bi-directional attention breaks the causality characteristics inherent to modern pre-trained LLMs, leading to potential performance degradation.
>
> Nevertheless, our D-Attn is flexible enough to incorporate bi-directional attention. Consider the scenario where we apply bi-directional attention to textual embeddings of question and visual embeddings while maintaining causal self-attention on textual embeddings of answer to not break causality during inference, the attention operation can be illustrated in the figure below:
>
> - [Bi-directional attention](https://imgur.com/bBGExep)
>
> Following our D-Attn framework, we can merge the info from blocks 1 and 2; blocks 3 and 4; blocks 5, 6, and 7; via our proposed alpha-weighting strategy. V2V Attn in block 1 can also be diagonalized. Compared with vanilla uni-directional D-Attn in our main paper, this design incurs extra computation in block 2 V2Q Attn with $\mathcal{O}(|V||Q|)$. Thus, the overall computational complexity is still linear with respect to |V|.
>
> ### **3. Visual embeddings as query**
> To the best of our knowledge, there aren't many successful applications of LVLM for vision-centric tasks such as visual search and face matching. Embedding learning [[1]](https://arxiv.org/abs/2105.14432)[[2]](https://arxiv.org/abs/2404.00262)[[3]](https://arxiv.org/abs/2401.17592)[[4]](https://arxiv.org/abs/2408.02766) may still be the SoTA in these scenarios.
>
> Nevertheless, it is possible to extend our D-Attn framework to support such scenario. Similar to the bi-directional attention described above, the attention between image1 $V$, image2 $U$, and text can be illustrated in the figure below:
>
> - [Multi-images](https://imgur.com/bUaJSYv)
>
> We use bi-directional attention between $V$ and $U$. Following our D-Attn framework, we can merge the info from blocks 1 and 2; blocks 3 and 4; blocks 5, 6, and 7; via our proposed alpha-weighting strategy. $V2V$-Attn in block 1 and $U2U$-Attn in block 4 can both be diagonalized. The interaction between $V$and $U$ happens in blocks 2 and 3, where one image acts as query to gather information from another image.
>
> ### **4. Dynamically assign query, key, and value**
> The idea of dynamically assigning query/key/value is very interesting and we did not see any LVLM literature implementing this idea. Our D-Attn architecture can be extended with a dynamic assignment module to support this scenario. Switching the role of query/key/value does not change the computation complexity within the LLM, still resulting in a linear complexity with respect to |V|. However, the additional dynamic assignment module may introduce extra computational cost.

---

> > ### Author Response · Authors · 2024-11-25
> >
> > Thank you for your valuable feedback on our submission. We have provided a detailed response to your comments and would appreciate any further clarifications or thoughts you may have. As the discussion period is ending soon, if there is anything else you would like us to address further, please let us know. We look forward to your input and greatly value your feedback.

---

> > > ### Comment · Reviewer_tnid · 2024-12-02
> > >
> > > Thank you for your detailed response and the comprehensive clarification of your approach. While your submission demonstrates impressive performance metrics, I maintain my initial assessment due to several structural considerations. Specifically, the separation of query and key components, while potentially beneficial for certain tasks, raises concerns about the model's generalization capabilities compared to traditional Transformer architectures.

---

### Author Response · Authors · 2024-12-02
**Summary of response**

We appreciate all the reviewers for their thoughtful and valuable feedback.

In this paper, we propose a new decomposed attention (D-Attn) architecture for LVLMs. When compared with conventional self-attention (S-Attn) architecture (e.g. LLaVA), our D-Attn model consistently boosts model performance while substantially reducing computational costs. The main idea is to process visual and textual embeddings differently, which is achieved by:
1. Decomposing self-attention on concatenated visual and textual embeddings into three separate attention operations: V2V, T2V, and T2T, and then merging them back with our proposed $\alpha$-weighting merging strategy. The decomposition gives us the flexibility to process visual and textual embeddings differently, while $\alpha$-weighting maintains the mathematical equivalence to self-attention, which is key to superior model performance as demonstrated in the ablation analyses.
2. Replacing the V2V Self-Attn with Diagonal-Attn/FCs so that the computational complexity is reduced from $\mathcal{O}(|V|^{2})$ to $\mathcal{O}(|V|)$.
3. Identifying the positional bias in T2V attention. We propose debiased positional encoding to remove this undesirable bias and boost model performance.

As suggested by the reviewers, we have thoroughly revised our manuscript and replied to each of the questions raised in the reviews:
- Reviewer **tnid**: We provide experiments and explanations for architectural variants, including textual embeddings placed before visual embeddings (Q1, Q3), bi-directional cross-modal attention (Q1, Q2), visual embeddings as query (Q2), potential information loss in V2V diagonal attention (Q5), and dynamically assigning query, key, and value (Q4).
- Reviewer **tS4a**: We provide the analyses of the proposed method's effectiveness and revise the manuscript accordingly (W2, W3, Q1, Q2). We provide explanations/clarifications for $\alpha$-weighting merging strategy and the disadvantages of other merging strategies (W1, Q3, Q4), as well as visual and textual modality misalignment (W3, Q2).
- Reviewer **1sRA**: We thoroughly discuss the novelty (W1, Q1) and motivation (W3, Q3) of this paper. We also provide additional analysis for $\alpha$-weighting merging strategy on various benchmarks (W2, Q2).
- Reviewer **KDE8**: We provide clarifications and experimental results for V2V Diagonal-Attn in the scenarios of single image and multi-images (W1), and the computational advantage of V2V Diagonal-Attn for high-resolution images (Q1). We also thoroughly explain the contribution of proposed $\alpha$-weighting (W2), visual and textual modality gap (W3), and relative position of text and image (Q2).

We once again express our heartfelt gratitude to all the reviewers for their valuable feedback, and we hope that our responses satisfactorily address all concerns. Please feel free to let us know if you have any remaining concerns and we are happy to address them!

---

### Meta-Review · Area_Chair_YUh4 · 2024-12-22

**Metareview:**

The paper proposes a new attention pattern for the visual tokens in a llava-like VLM. In particular, the paper introduces D-Attn, which allows for only self-attention on the visual tokens, as well as an alpha-weighting scheme that merges visual and textual information. The experiments build on the llava architecture and show speed up due to the diagonalisation in the visual token's attention pattern, while maintaining performances. However, the paper currently has several weaknesses such as in methodology (diag V2V attention is simply an MLP and not attention), specialised patterns being necessary for multi-image cases, lack of thorough analysis of the alpha-weighting's internals, and the unclear motivation for the method. Ultimately the AC also does not find it clear what is the exact contribution: efficiency, alpha weighting, D-Attn, which could be refined if the motivation is more clear. As such the AC recommends rejection.

**Additional Comments On Reviewer Discussion:**

There was good discussion between the reviewers and the authors. The reviewer tnid engaged in the discussion but ultimately the limitation of the current paper's approach inflexibility of image to text attention to be a remaining weakness. tS4a did raise initial points that were addresses in the discussion and raised their score to marginally above acceptance. Similarly, 1sRA's points got cleared, but also maintains that issues with the motivation of the method remain, Lastly, reviewer KDE8 engaged in extensive discussions with the authors, who maintains the lack of novelty of the alpha-weighting component and modality gap relevance.
Overall this paper has become substantially more refined during the discussion period but the points above remain and put it just below the high bar of acceptance for ICLR. The AC hopes the authors take the points and in particular refine the motivation for D_Attn further and resubmit to another venue.

---

### Decision · Program_Chairs · 2025-01-22

Reject